# Colorectal Cancer Cell Invasion and Functional Properties Depend on Peri-Tumoral Extracellular Matrix

**DOI:** 10.3390/biomedicines11071788

**Published:** 2023-06-22

**Authors:** Marco Franchi, Konstantinos-Athanasios Karamanos, Concettina Cappadone, Natalia Calonghi, Nicola Greco, Leonardo Franchi, Maurizio Onisto, Valentina Masola

**Affiliations:** 1Department for Life Quality Studies, University of Bologna, 47900 Rimini, Italy; 2Department of Pharmacy and Industrial Pharmacy, University of Bologna, 40126 Bologna, Italy; konstantin.karamanos@studio.unibo.it; 3Department of Pharmacy and Biotechnologies, University of Bologna, 40126 Bologna, Italy; concettina.cappadone@unibo.it (C.C.); natalia.calonghi@unibo.it (N.C.); 4Department of Biomedical Sciences, University of Padova, 35122 Padova, Italy; nicola.greco@phd.unipd.it (N.G.); maurizio.onisto@unipd.it (M.O.); valentina.masola@unipd.it (V.M.); 5Department of Medicine, University of Bologna, 40126 Bologna, Italy; leonardo.franchi@studio.unibo.it

**Keywords:** colorectal cancer, doxorubicin, matrigel, type I collagen, matrix metalloproteinases, 3D cell cultures, scanning electron microscopy, epithelial-to-mesenchymal transition (EMT)

## Abstract

We investigated how the extracellular matrix (ECM) affects LoVo colorectal cancer cells behavior during a spatiotemporal invasion. Epithelial-to-mesenchymal transition (EMT) markers, matrix-degrading enzymes, and morphological phenotypes expressed by LoVo-S (doxorubicin-sensitive) and higher aggressive LoVo-R (doxorubicin-resistant) were evaluated in cells cultured for 3 and 24 h on Millipore filters covered by Matrigel, mimicking the basement membrane, or type I Collagen reproducing a desmoplastic lamina propria. EMT and invasiveness were investigated with RT-qPCR, Western blot, and scanning electron microscopy. As time went by, most gene expressions decreased, but in type I Collagen samples, a strong reduction and high increase in MMP-2 expression in LoVo-S and -R cells occurred, respectively. These data were confirmed by the development of an epithelial morphological phenotype in LoVo-S and invading phenotypes with invadopodia in LoVo-R cells as well as by protein-level analysis. We suggest that the duration of culturing and type of substrate influence the morphological phenotype and aggressiveness of both these cell types differently. In particular, the type I collagen meshwork, consisting of large fibrils confining inter fibrillar micropores, affects the two cell types differently. It attenuates drug-sensitive LoVo-S cell aggressiveness but improves a proteolytic invasion in drug-resistant LoVo-R cells as time goes by. Experimental studies on CRC cells should examine the peri-tumoral ECM components, as well as the dynamic physical conditions of TME, which affect the behavior and aggressiveness of both drug-sensitive and drug-resistant LoVo cells differently.

## 1. Introduction

Colorectal cancer (CRC) is the third leading cause of cancer-related death in the world, the third most common cancer in men, and the second in women worldwide [1]. Despite the success obtained in CRC treatments including surgery, radiation therapy, and chemotherapy, many patients stop responding to the selected drugs. Failure of chemotherapy, which develops during CRC drug treatment, still represents a dramatic problem in oncology [2,3]. One of the many anticancer drugs involved in chemoresistance is doxorubicin, which acting by interfering with the function of DNA is used in solid and liquid tumors, as well as in CRCs. Both intrinsic or acquired chemoresistance lead to cancer recurrence, which is associated with a poor prognosis and shorter survival of the patient. Among the epigenetic factors favoring chemoresistance, many natural components of the extracellular matrix (ECM) can play an important role in driving the tumor microenvironment (TME) remodeling, which may favor therapeutic failure [3].

Healthy extracellular matrices (ECMs) act as dynamic mixtures of different molecules secreted by stromal cells and provide complex structural and chemical support to tissues and organs by allowing mechanical and molecular crosstalk between epithelial and stromal cells [4]. Different research confirms that similar interplay also occurs between cancer and the ECM of the tumor microenvironment (TME) and suggests that no tumor can invade if no structural and molecular changes develop in the peri-tumoral ECM [5,6,7,8]. The biomolecular and physical changes of the tumor microenvironment (TME) represent a complex response from host tissues to limit or avoid cancer cell invasion. However, when cancer cells can cross the natural barriers of ECM, are not hindered by TME changes, or furthermore, can induce peri-tumoral ECM modifications favoring cell migration into deep tissues, they are considered malignant cancer cells which become able to colonize distant organs.

Mechanical properties such as stiffness, density, microporosity, fluidity, or viscoelasticity of ECMs regulate cell migration and cancer cell invasion [9,10,11,12,13,14]. Structural TME remodeling changes the physical properties of the peri-tumoral tissue. In breast cancer, the progressive development of the tumor matrix collagen array has been classified in a tumor-associated collagen signature classification (TACS), which distinguishes the different orientations of the peri-tumoral collagen fibers [15,16,17,18,19,20,21]. Among the sequential changes of the collagen fibers array, the first host response and adjustment concerning the TME in most solid tumors consist of the deposition of type I collagen fibrils, which forms a reinforced fibrotic barrier around the tumor mass [22,23]. The mechanical properties of the peri-tumoral ECM, and in particular the desmoplastic tissue, can influence tumor development and cancer cell behavior, while also promoting drug treatment resistance. In particular, the fibrotic tissue surrounding the tumor mass seems to physically limit the entry of immune cells and drugs so as to reduce the therapeutic capability of anticancer therapies and induce chemoresistance [24,25].

The first dynamic event, which then allows CRC cells’ intravasation by penetrating blood or lymphatic vessels and metastasis by colonizing distant organs, occurs when the cells cross the basement membrane (BM) and move to invade the ECM of lamina propria of the large bowel. Many in vitro studies investigated cancer cells in 2D cultures, underestimating the role of free cell movement and duration of culturing cues in regulating cancer invasion. In a previous study, we demonstrated that in 3D cultures, whose Millipore filters did not allow cells to cross the filters, the low aggressive and sensitive-to-doxorubicin LoVo-S CRC cells developed an EMT phenotype and became more aggressive when they grew in contact with concentrated type I collagen. In contrast, the more aggressive and resistant to doxorubicin LoVo-R CRC cells did not change their morphological phenotype or epithelial to mesenchymal transition (EMT) markers’ expression in different concentrations of different ECM substrates. [26]. With the aim to better understand the dynamic behavior of LoVo-S and LoVo-R CRC cell invasion of the ECM, in this study we investigated the behavior and phenotype of these cancer cells when they were free to move and invade Matrigel, mimicking the BM or type I collagen meshwork, reproducing the collagen array of the bowel desmoplastic lamina propria.

## 2. Materials and Methods

### 2.1. Cell Cultures

Human CRC cell line LoVo-S and its drug-resistant subline LoVo-R, which was obtained in vitro after repeated expositions of LoVo cells to 1 µg/mL of doxorubicin, were cultured in the RPMI 1640 medium, with the addition of 10% fetal bovine serum (FBS), 2 mM L-glutamine, penicillin (100 U/mL), and streptomycin (100 µg/mL), at 37 °C with 5% CO2 in humidified air. Resistance to doxorubicin was verified before each experiment, as previously reported [27,28]. When cells reached 80% confluence, they were detached with the Trypsin-EDTA solution (1.5 × 10^5^) and were seeded for 3 and 24 h on “Isopore Membrane Filters” with a pore size of 8.0 µm (Millipore, Milan, Italy) previously covered with Matrigel (BD Biosciences, Milan, Italy) or type I Collagen solution (C3867, Sigma-Aldrich, Schnelldorf, Germany) mimicking structural ECM natural barriers. Matrigel and Collagen filter coatings were prepared by diluting Matrigel and type I Collagen at the proper concentrations (0.2 or 3.5 mg/mL, respectively) in sterile water (pH 6), dispersed on the filters, and incubated at 37 °C for 2 h for polymerization. The lower chamber was filled with F-12K 20% FBS. We decided to use Isopore Membrane Filters with a pore size of 8.0 µm, which allowed any cell to cross through and granted us the ability to analyze LoVo-S/-R cells during ECM invasion.

### 2.2. RNA Isolation and Real-Time qPCR Analysis

We extracted total RNA from cells with a Trizol reagent (Invitrogen, Thermo Fisher, Waltham, MA, USA) following the manufacturer’s instructions [29]. We used a Nanodrop spectrophotometer (EuroClone, Milan, Italy) to check RNA yield and purity, and MoloneyMurine Leukemia Virus Reverse Transcriptase (Sigma-Aldrich) to transcribe the total RNA into cDNA from each sample. Using SensiFAST SYBR Hi-Rox (Bioline, LABGENE SCIENTIFIC SAZI, Châtel-Saint-Denis, Switzerland), we performed Real-time PCR on a StepOne™ Real-Time PCR System (Thermo Fisher, Waltham, MA, USA). To quantify gene expression, we used the comparative Ct method (DDCt), and the relative quantification was estimated as 2-∆∆Ct. Melting curve analysis excluded the presence of non-specific amplification products. The forward and reverse primer sequences are shown in Table 1.

### 2.3. E-Cadherin Expression by Western Blot

LoVo-S and LoVo-R cells were seeded (1.5 × 10^5^ cells/cm^2^) on Millipore filters covered with Matrigel (0.2 mg/mL) or type I Collagen (3.5 mg/mL) to evaluate protein expression. We removed the medium after 3 and 24 h, washed the cells with PBS, and lysed them in RIPA buffer composed of 50 mM Tris–HCl, pH 5.0, 150 mM NaCl, and 0.5% Triton X-100 with the Complete Protease Inhibitor Mixture (Roche Applied Science, Enzberg, Germany). Equal amounts of proteins were treated with reducing sample buffer and denatured for 10 min at 100 °C after quantification. The protein samples were resolved in 10% SDS–PAGE and were then electrotransferred to cellulose membranes. Non-specific binding was blocked for 1 h at room temperature with non-fat milk (5%) in TBST buffer (50 mMTris–HCl, pH 7.4, 150 mM NaCl, and 0.1% Tween 20). The filter membranes were exposed to primary antibodies (1:1000) directed against GAPDH (sc-47778 Santa Cruz) and E-CADHERIN (E-CAD) (GTX10443 GeneTex, Irvine, CA, USA) overnight at 4 °C and incubated for 1 h at room temperature with a secondary peroxidase-conjugated antibody. The signal, detected with Luminata™ Forte Western HRP Substrate (Millipore) according to the manufacturer’s instructions, was successfully acquired with Mini HD9 (UVItec, Cambridge, UK).

### 2.4. MMPs Activity by Zymography

LoVo-S and LoVo-R cells were seeded (1.5 × 10^5^ cells/cm^2^) on Millipore filters covered with Matrigel (0.2 mg/mL) or type I Collagen (3.5 mg/mL) to evaluate MMPs activity. We removed the medium after 3 and 24 h, washed the cells with PBS, and lysed them in RIPA buffer without Protease Inhibitor. We resolved equal amounts of protein in a non-reducing sample buffer on 10% SDS-polyacrylamide gels co-polymerized with 0.1% gelatin. After electrophoresis, to remove SDS, the gels were washed twice for 30 min in 2.5% Triton X-100 at room temperature. Then they were equilibrated for 30 min in collagenase buffer to be finally incubated overnight with fresh collagenase buffer at 37 °C. Gels were stained for 1 h in 0.1% Coomassie Brilliant Blue R-250 and 30% MeOH/10% acetic acid after incubation and destained in 30% MetOH/10% acetic acid. Digestion bands were analyzed.

### 2.5. Statistical Analysis

We performed Statistical analyses on real-time PCR data using the Relative Expression Software Tool (REST). To compare the two distributions, we used the two-tailed *t*-test. For multiple comparisons, one-way analyses of variance (ANOVA) were used with Sidak’s test (for multiple comparisons) using GraphPad Prism. *p* < 0.055 was considered significant for all tests.

### 2.6. Scanning Electron Microscopy

3D cultures of LoVo-S and LoVo-R CRC cells seeded on isopore membrane filters (Millipore, Milan, Italy) with 8 µm pore size, covered with different biological substrates (Matrigel or type I Collagen) at different concentrations (0.2 or 3.5 mg/mL), were performed. After 3 and 24 h of culturing, all samples were completely immersed in Karnovsky’s solution fixative for 20 min at 4 °C. We rinsed the samples three times with 0.1% cacodylate buffer, and gradually dehydrated them with increasing concentrations of ethanol and final hexamethyldisilazane (Sigma-Aldrich, Inc., Burlington, MA, USA) for 15 min. Once mounted on proper stubs, the specimens were coated with a 5 nm palladium gold film (Emitech 550 sputter-coater, Quorum Technologies, Lewes, UK) and then observed under a scanning electron microscope (SEM) (Philips 515, Eindhoven, The Netherlands) operating in secondary-electron mode.

## 3. Results

### 3.1. Evaluation of EMT Markers and Matrix Degrading Enzymes in CRC Cells Cultured in Different Matrix Substrates

We first tested the expression of the EMT-related genes E-cadherin, vimentin, and Snail to evaluate if the different matrix substrates influence the behavior of the two CRC cell types, LoVo-S and LoVo-R, when they cross the BM or invade a desmoplastic lamina propria. This was performed by total RNA isolation and RT-qPCR analysis of the CRC cells cultured for 3 and 24 h on a Millipore filter allowing cell migration, which was covered by Matrigel or type I collagen. The epithelial marker E-cadherin exhibited a much higher expression in LoVo-S as compared to LoVo-R cells in both Matrigel and type I Collagen substrate groups after 3 h. However, in the 24 h groups, E-cadherin expression increased in LoVo-S cells cultivated on Matrigel and in LoVo-R cells cultivated on Collagen (Figure 1A). The expression of the mesenchymal marker vimentin was significantly higher in LoVo-R as compared to LoVo-S cells in all substrates both after 3 and 24 h. As time went by, vimentin expression increased in LoVo-S cells cultured on both Matrigel and type I Collagen but did not change in LoVo-R cells (Figure 1B). Similarly, the mesenchymal marker SNAIL was also significantly more expressed in LoVo-R as compared to LoVo-S cells in all substrates. As time went by, it decreased in all groups from 3 to 24 h of culturing (Figure 1C). In general, these data suggest that LoVo-S cells are less aggressive than LoVo-R cells, and the duration of culturing can affect E-cadherin and SNAIL expression in LoVo-R cells cultivated on type I Collagen.

We also evaluated the expression of matrix-degrading enzymes such as metalloproteinase-2, -9, and -14 (MMPs) and heparanase (HPSE) to evaluate the capability of LoVo cells in degrading the ECM during invasion. In general, with the same duration of culturing, all the MMPs were more expressed in Type I Collagen cultures than in Matrigel cultures, both in LoVo-S and LoVo-R cells. However, the MMP expression decreased in both cell types cultivated in all the substrates after 24 vs. 3 h, with the exception of MMP-2, which strongly increased in LoVo-R cells on type I Collagen as time went by (Figure 2A–C). HPSE was more expressed in LoVo-R vs. LoVo-S cells in all the substrates after 3 h, and was absolutely higher in LoVo-R cells on Matrigel. After 24 h, it decreased in LoVo-R cells cultivated on all the substrates but increased in LoVo-S cells on Matrigel (Figure 2D). Both culturing durations and the type of substrate seem to differently affect MMP expression, as also confirmed by the protein level WB and zymography (Figure 3).

### 3.2. Ultrastructural Morphological Features of LoVo-S/-R CRC Cells Cultured on Millipore Covered by Matrigel Mimicking the BM or Type I Collagen Mimicking the Desmoplastic Lamina Propria after 3 h

To evaluate the ultrastructural phenotype of LoVo-S/-R CRC cells growing on and crossing a Matrigel layer mimicking the BM, we used SEM analysis. In particular, Matrigel covered Millipore filters with an 8 µm pore size, which allowed cell invasion and the examination of the cells when they began to cross the filter barrier after 3 h of culturing.

LoVo-S cells cultured on the Millipore filter covered by Matrigel (0.2 mg/mL) appeared grouped, and all of them showed cell–cell contacts. To properly explore the surrounding microenvironment, they developed long filopodia and intercellular tunneling nanotubes. Cells primarily showed a globular shape developing cytoplasmic extravesicles, which, for their size, were identified as exosomes and microvesicles. However, very few elongated, mesenchymal-shaped cells were also detectable (Figure 4A–C).

When the LoVo-S cells were cultivated on high-concentrated type I Collagen (3.5 mg/mL), mimicking the collagen array of the desmoplastic lamina propria, they appeared less grouped than the same cultivated on Matrigel. They showed both equally distributed elongated-mesenchymal phenotypes, developing both filopodia or lamellipodia, and globular shapes (Figure 4D). The surface of the cells, which firmly adhered to the collagen fibril surface, appeared very smooth with no extravesicles (Figure 4E,F). Fibrils formed a collagen meshwork with very small (less than 1 µm) inter fibrillar spaces or pores (Figure 4F).

LoVo-R cells growing on Matrigel appeared more isolated than the LoVo-S cells and showed a globular shape with more evident exosomes and microvesicles on their surface. Only a few elongated-fusiform mesenchymal-shaped cells were observed (Figure 5A–C). At higher magnifications, the globular-shaped cells exhibited filopodia and lamellipodia adhering to the Matrigel layer (Figure 5B,C). Some globular-shaped cells were observed while crossing the pores of the Millipore filter (Figure 5B,C).

The LoVo-R cells cultivated on type I Collagen mimicking the collagen array of the desmoplastic lamina propria exhibited a globular shape with exosomes and microvesicles on their surface (Figure 5D,E). The cells appeared as single isolated cells, but upon adhering to the collagen fibrils they developed cytoplasmic protrusions or short filopodia, which seemed to penetrate into the collagen layer (Figure 5F).

### 3.3. Ultrastructural Morphological Features of LoVo-S/-R CRC Cells Cultured on Millipore Covered by Matrigel Mimicking the Basement Membrane or Type I Collagen Mimicking the Collagen Network of Lamina Propria after 24 h

We analyzed the ultrastructural phenotype of LoVo-S/-R CRC cells growing on and crossing the Matrigel covering a Millipore filter using SEM after 24 h. The LoVo-S cells appeared as grouped and flattened epithelial cells exhibiting tight contact with each other and surrounded by a fibrillar meshwork, likely corresponding to remnants of the Matrigel substrate (Figure 6A,B). Clusters of several grouped LoVo-S cells showed a leader cell, which developed short filopodia and drove the other cells to collectively cross the pores of the Millipore filter (Figure 6C).

The LoVo-S cells growing on type I Collagen developed an epithelial-like phenotype. They appeared as polygonal and flattened cells with tight cell–cell contact. (Figure 6D). On this continuous layer of flattened cells, few isolated cells exhibited a funnel-shaped phenotype (Figure 6E). Some of these cells invaded the Matrigel and penetrated into the pores of the Millipore filter (Figure 6F).

The LoVo-R cells cultivated for 24 h on the Matrigel layer appeared as grouped epithelial cells with cell–cell contact but developed more microvilli and extravesicles on their surface compared to the LoVo-S cells (Figure 7A). Remnants of Matrigel enveloped the LoVo-R cells (Figure 7B). Single cells and clusters of grouped LoVo-R cells driven by a leader cell developed filopodia to cross the pores of the Millipore filter (Figure 7C,D).

When the LoVo-R cells were cultivated on a type I Collagen meshwork, they developed a funnel-shaped phenotype but showed weak cell-to-cell contact through thin and short filopodia (Figure 7E,F). Some cells developing extravesicles completely invaginated the collagen fibril layer (Figure 7G) and developed from their ventral side short and thin cytoplasmic protrusions which adhered to the fibrils and morphologically corresponded to invadopodia (Figure 7H).

## 4. Discussion

Solid tumors interplay with their TME, which regulates the EMT and invasion capability of cancer cells but can also play a primary role in favoring the dramatic development of drug resistance. Among the different factors favoring chemoresistance in cancer cells, the dense fibrotic tissue developing around the tumor mass seems to act as a physical barrier limiting the entry of immune cells and drugs, so as to reduce the host response and the effect of anticancer therapies [24,25]. Changes in the ECM array and composition occurring in the TME could be primarily related to CAFs, which alter the collagen array and composition of the ECM surrounding the tumor mass and have been considered capable of indirectly regulating both cancer fate and therapeutic success [15,17,18,19,20,21,24,25,30]. Therefore, it is crucial to understand how drug-resistant cancer cells invade the TME to better explain how the peri-tumoral ECM can affect their invasion.

The relationship between cancer cell invasiveness and TME is particularly focused on cell movement, which is strongly related to the gene expression of EMT markers and matrix effectors, as well as the morphological phenotype. Many different MMPs act as modulators of the TME by degrading proteins in the ECM, but MMP-2, -9, and -14 are particularly involved in cancer cell invasion of many tumors [31]. They are secreted by exocytosis from cytoplasmic protrusions called invadopodia but may also be released in the TME from cytoplasmic extravesicles shed by a superficial blebbing of the plasma membrane. Extravesicles include multivesicular bodies or exosomes (50–200 nm), vesicles deriving from direct budding of the plasma membranes or microvesicles (50–1000 nm), and larger vesicles or apoptotic bodies (>1000 nm). They contain lipids, second messengers, genetic material, and HPSE [32,33]. Recently, live confocal imaging has been applied both in vitro and in vivo to evaluate the “dynamic behaviors” of invading cancer cells in the ECM [34,35,36]. However, this fascinating technique has limitations in obtaining high magnification and resolution, which allow one to observe changes in cancer cell phenotypes and the development of cytoplasmic protrusions related to ultrastructural changes of the TME. Following these considerations, we planned a project which using SEM investigation compared to EMT marker and matrix effector expressions, tries to explain the spatiotemporal effect of the ECM components on LoVo-R (resistant to doxorubicin) vs. LoVo-S (sensitive to doxorubicin) CRC cells. On its own, doxorubicin resistance induces EMT, the downregulation of E-cadherin, and an increase in the expression of mesenchymal markers such as N-cadherin and vimentin in LoVo-R vs. LoVo-S cells [37]. In a previously published study, we first analyzed these cells when they were confined to the ECM barriers and had no possibility to freely move. In particular, we investigated the behavior of LoVo-S and LoVo-R cells growing in ECM substrates covering Millipore filters, which did not mechanically allow them to invade after 24 h. We found that when low aggressive LoVo-S cells were hindered to cross the Millipore filter and only tried to invade a concentrated type I Collagen substrate (3.5 mg/mL) they developed a morphological and biomolecular EMT phenotype. Surprisingly, the more aggressive doxorubicin-resistant LoVo-R cells appeared independent on TME, because they did not change their morphological phenotype, EMT markers, and HPSE expression in all the different tested ECM substrates [26].

In this study, we proceeded to investigate the LoVo CRC cells when they were free to move and play a spatiotemporal invasive journey into the natural barriers of TME ECM. Therefore, LoVo-S and LoVo-R CRC cells were cultivated in 3D cultures for 3 and 24 h using Millipore filters as a scaffold, whose pore size (8 µm) allowed cells to pass through and invade Matrigel or type I Collagen, mimicking the BM and collagen array of the desmoplatic lamina propria, respectively. We observed that both the culturing duration and type of substrate can affect LoVo-S and LoVo-R cells’ behavior differently when the cells can freely move and invade the ECM substrates. The Matrigel layer acted as a valid biological barrier inducing a decrease in all EMT markers and matrix effectors in both LoVo-S and -R cells over time, whereas the type I Collagen substrate affected the two cell types differently as time went by. In general, type I Collagen induced an increase in all MMPs expression vs. the Matrigel substrate in LoVo-S cells after both 3 and 24 h, but the gene expression of most EMT markers and matrix effectors decreased over time, as confirmed by the protein-level analysis and the development of a mesenchymal-to-epithelial transition (MTE) morphological phenotype. Interestingly, the strong decrease in MMP-2 expression (ca. 20.5 times) observed in LoVo-S cells was dependent on time, but it was also influenced by the type I Collagen substrate when compared to the corresponding Matrigel samples. Moreover, in LoVo-R cells, MMP-9, -14, and HPSE expression decreased over time. However, a clear increase in the MMP-2 expression (ca. 2.8 times) in these cells cultured on type I Collagen occurred over time. This increase was particularly related to the type of substrate (collagen) if we consider that MMP-2 expression decreased in the same cells cultivated in Matrigel (Figure 8). These data were supported by the development of invaginating and invading morphological phenotypes showing invadopodia in contact with the underlying collagen fibrils after 24 h, as well as by the protein-level analysis. Our interest was particularly focused on the highly concentrated type I collagen, which was just reproduced to carefully mimic the collagen meshwork in the desmoplastic lamina propria of the colon wall. Interestingly, the ultrastructural analysis of this natural barrier showed that it consists of fibrils showing a relatively large diameter and creating inter fibrillar micropores. The size and stiffness of type I Collagen fibrils, as well as the particular microporosity of the collagen meshwork, might induce the proteolytic invasion of the drug-resistant LoVo-R cells into the deep ECM of the TME. Furthermore, our results demonstrate that the physical array of a natural substrate such as the collagen meshwork can regulate the behavior of LoVo subtype cells. As time passes, type I Collagen can reduce the invasiveness and aggressiveness of drug-sensitive LoVo-S cells, but at the same, improves the proteolytic aggressiveness of chemoresistant LoVo-R cells.

The decrease over time in the E-cadherin expression in LoVo-S cells cultivated on type I Collagen and the increase in the same enzyme in LoVo-R cells in similar cultures apparently seem in contrast with our previous considerations. However, it was reported that the conservation of E-cadherin expression for intercellular links contributes to collective migration. Indeed, some tumor cell clusters maintain their epithelial phenotype and are more effective at forming metastases than single cells [38,39,40].

Data obtained after 24 h from these free-migrating cells were compared with the results reported in a previous study, where LoVo cells were cultivated for 24 h on the same ECM substrates covering a Millipore filter whose pore size stopped cell migration [26]. After 24 h, the freedom of movement attenuated LoVo-S cells’ aggressiveness by reducing all the MMPs in both LoVo-S and -R cells cultivated in any substrate. In particular, the stronger decrease in MMP-2 and -9 expression in type I Collagen cultures vs. the corresponding Matrigel samples suggested that the type I Collagen substrate plays a primary role in reducing LoVo-S cells’ aggressiveness. In contrast, type I Collagen induced a strong increase in MMP-2 expression (ca. 7.5 times), which was also related to the type of substrate, as MMP-2 expression in Matrigel cultures decreased over time (Figure 9). These results suggest that when chemoresistant LoVo-R cells invade and cross the sub-epithelial collagen environment, they improve their dangerous proteolytic capability.

Our data concerning cancer cell aggressiveness and MMP expression are supported by different studies. MMP-2 and -9 are dramatically involved in BM invasion since both are able to cleave gelatin and type IV collagen. However, it was reported that MMP-2 can also digest normal and denatured type I collagen [41,42,43]. In colorectal cancer, as well as in other solid tumors, both the plasma MMP-2 and -9 levels are reported to be related to clinical staging and could potentially represent an indicator of invasion or metastasis [44,45,46,47]. However, some researchers reported that MMP-2 is highly prognostic for colorectal cancer survival vs. MMP-9 as it is significantly increased in patients with lymph node metastasis compared to those without [46,47,48,49]. Moreover, both the high expression of MMP-2 and low MMP-9 expression in both tumor and stroma cells seem to be associated with poor prognosis in cervical cancer, as well as in several tumors [50].

Even though we demonstrated that the duration of culturing, free cell migration, and the type of substrate differently affect LoVo-S and -R CRC cells’ behavior, the way that type I collagen improves proteolytic aggressiveness in invading LoVo-R cells over time is still unclear. Cancer cell migration and invasion modalities are expressions of cancer cell adaptation, and the different mechanisms of cancer cell movement including ameboid migration, mesenchymal migration, and collective cell migration are related to well-defined morphological phenotypes, which can be better analyzed via SEM [51]. Filopodia or lamellipodia of LoVo CRC cancer cells may be able to sense the TME physical array by exploring the free spaces, filled with fluids, along the route they want to move. This agrees with the proposed sensing capability of the epithelial cells, which can transmit different mechanochemical stimuli and adapt their behavior by reshaping their morphology [52]. Epithelial cells physically sense both mechanical forces and chemical changes in the ECM microenvironment via adhesion complexes such as focal adhesions and cortical actin networks activated by cell plasma membrane stimuli or changes in cell shape. Moreover, intercellular force transmissions are transferred by intercellular adhesions such as the adherens junctions, tight junctions, and desmosomes [53,54]. Recently, it was reported that cancer cells can sense hydraulic pressure in the extracellular fluids by membrane ruffling and make directional choices of migration by moving toward channels with higher viscoelasticity [55,56,57]. Similarly, LoVo-S and LoVo-R CRC cells, in both static and migrating conditions, could sense the physical features of TME and adapt to invade the different ECM substrates and cross the Millipore filter. Although this study reproduced the natural biological barriers of the bowel ECM in 3D cultures, our in vitro results need to be compared and discussed with clinical data, which will help in understanding how the ECM of the TME can drive the invasion of drug-resistant cancer cells.

## 5. Conclusions

Chemoresistance is a response of cancer cells to a therapy-induced hostile environment and promotes tumor progression. Both clinical and experimental studies suggest that the deposition of type I Collagen in the TME can hinder the diffusion of drugs and blocks the access of immune cells to the tumor, thus contributing to the development of chemoresistant cancer cells [24,25,58,59,60]. Both the duration of culturing and the free movement of cells differently influence the morphological phenotype and aggressiveness features of drug-sensitive LoVo-S and doxorubicin-resistant LoVo-R CRC cells cultured on a Millipore filter covered with different ECM substrates. In particular, the type I Collagen meshwork, mimicking a desmoplastic lamina propria, is the ECM substrate that more strongly affects these CRC cells in a spatiotemporal invasion. It attenuates the low aggressiveness in LoVo-S cells as confirmed by the development of epithelial-shaped cells, whereas it highly improves the proteolytic invasive capability of LoVo-R cells by promoting a strong increase in MMP-2 expression and the development of EMT phenotypes. Understanding how some ECM components affect drug-resistant CRC cells will help upcoming therapeutic procedures aiming to modify the altered ECM of the TME.

## Figures and Tables

**Figure 1 biomedicines-11-01788-f001:**
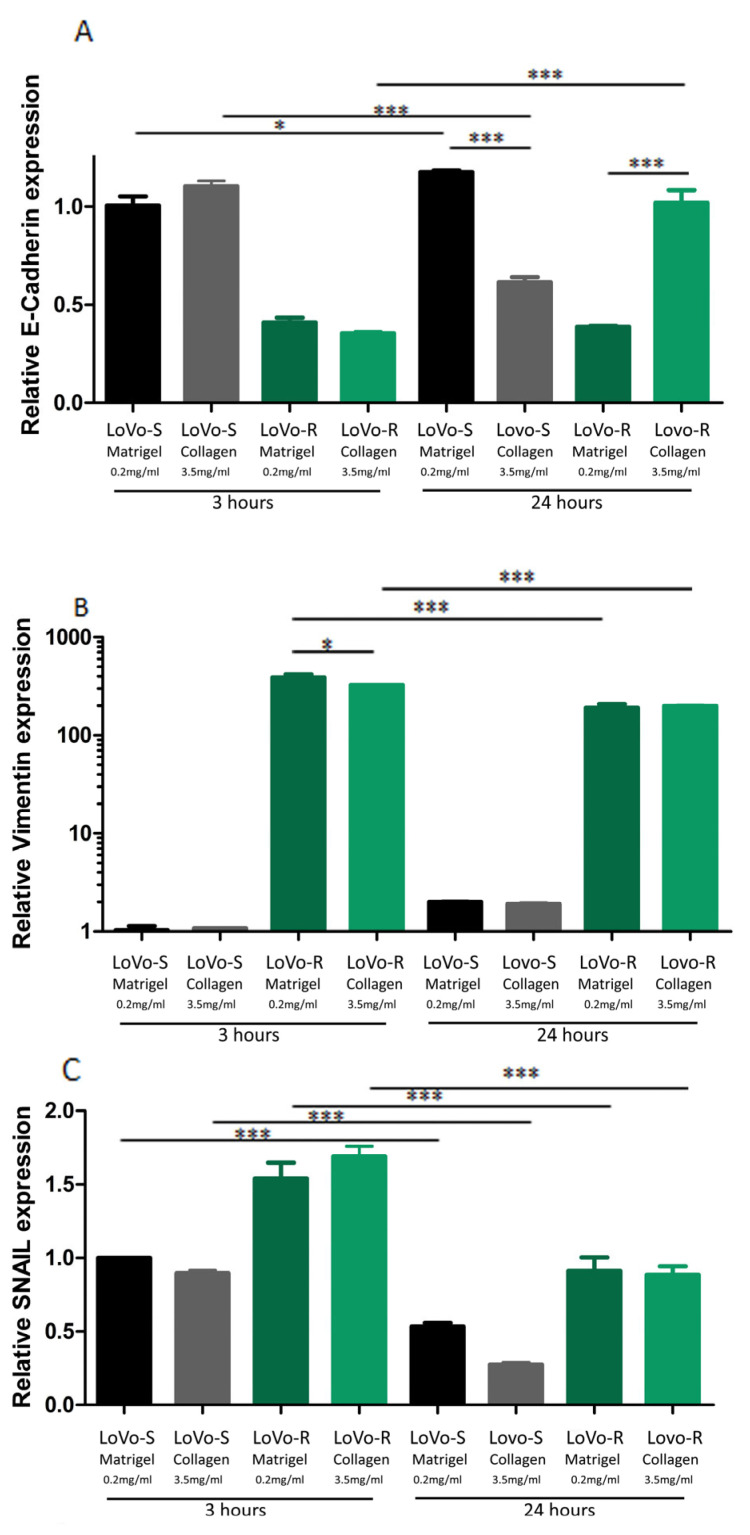
Evaluation of EMT Markers in CRC Cells Cultured in Different Matrix Substrates after 3 and 24 h. The gene expression of E-cadherin (**A**), Vimentin (**B**), and Snail (**C**) in LoVo-S cells cultured on Matrigel or type I Collagen after 3 and 24 h was evaluated by Real-Time PCR and normalized at GAPDH as housekeeping. Graphs represent Mean  ±  S.D. *n* = 6. * *p*  <  0.05, *** *p* < 0.0001.

**Figure 2 biomedicines-11-01788-f002:**
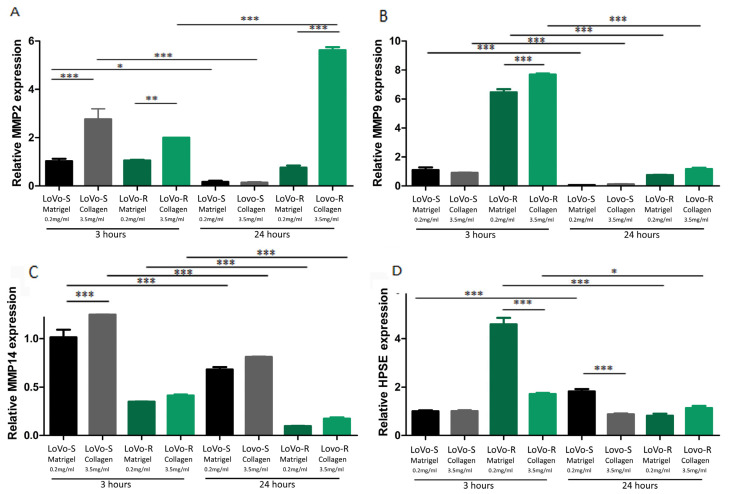
Evaluation of Matrix-Degrading Enzymes’ gene expression in LoVo CRC cells cultured in different matrix substrates after 3 and 24 h. The gene expression of MMP-2 (**A**), MMP-9 (**B**), MMp-14 (**C**), and HPSE (**D**) in LoVo cells cultured on Matrigel and type I Collagen for 3 and 24 h was evaluated by Real-Time PCR and normalized at GAPDH as housekeeping. Graphs represent Mean  ±  S.D. *n* = 6. * *p*  <  0.05, ** *p*  <  0.001, *** *p* < 0.0001.

**Figure 3 biomedicines-11-01788-f003:**
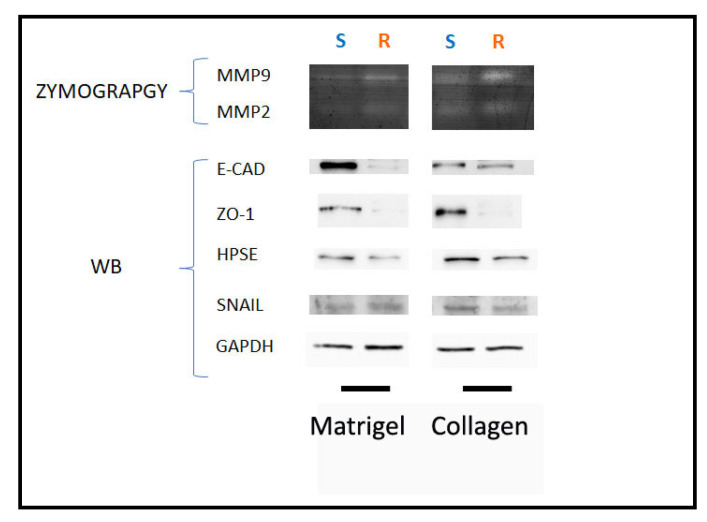
Evaluation of Matrix-Degrading Enzymes’ protein expression and activity in LoVo CRC cells cultured in different matrix substrates after 3 and 24 h. Upper: gelatin zymography shows MMP-9 and MMP-2 activity bands in protein extract of LoVo cells cultured on Matrigel and type I Collagen for 3 and 24 h. Lower: The protein expression of E-CAD, ZO-1, HPSE, and SNAIL was evaluated by Western Blot analysis. GAPDH was included as loading control.

**Figure 4 biomedicines-11-01788-f004:**
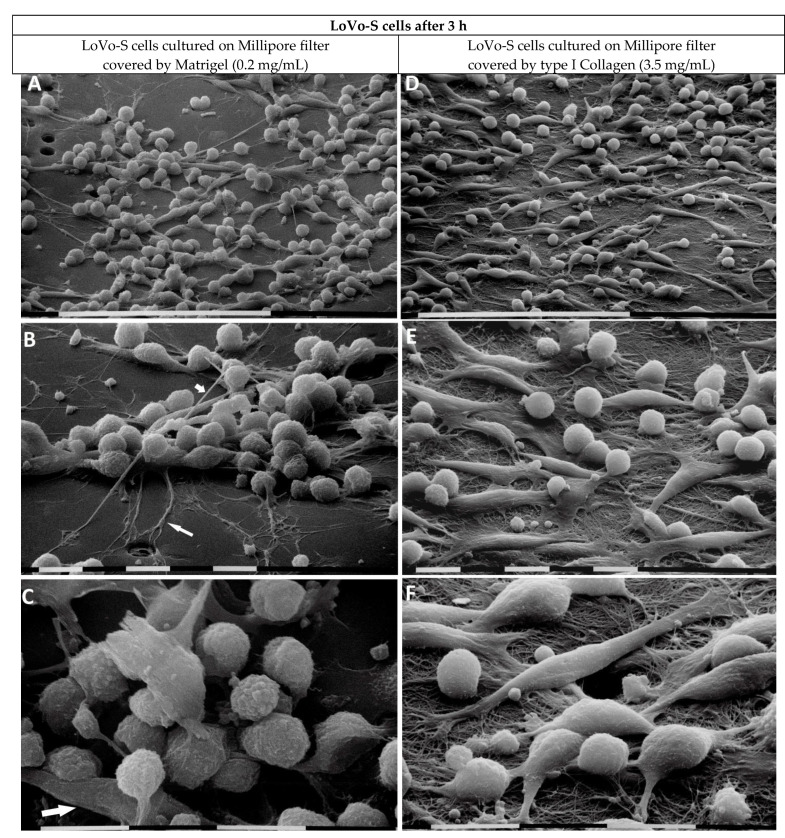
LoVo-S cells cultivated for 3 h on Millipore filter with 8 µm pores covered by Matrigel (0.2 µg/µL) mimicking the BM or type I Collagen (3.5 µg/µL) mimicking the collagen arrangement of the desmoplastic lamina propria. LoVo-S cells cultured on Matrigel (0.2 mg/mL) cover most of the Millipore pores. The cells appear grouped, and all of them show tight cell–cell contacts. Bar = 100 µm (**A**). Long filopodia (long arrow) for exploring the microenvironment, as well as intercellular tunneling nanotubes (short arrow) for cell interplay, are visible. Bar = 10 µm (**B**). The cells show a globular shape with exosomes and microvesicles on their cytoplasmic surface, but very few elongated-mesenchymal-shaped cells are also present (arrow). Bar = 10 µm (**C**). LoVo-S cells cultivated on highly concentrated type I Collagen (3.5 mg/mL) appear less grouped than the same cultivated on Matrigel. Bar = 100 µm (**D**). Both elongated-mesenchymal phenotypes, developing filopodia or lamellipodia, and globular-shaped cells are visible. Bar = 10 µm (**E**). The cells firmly adhere to the collagen fibril surface and most of them appear very smooth with no extravesicles. Single fibrils form a collagen meshwork with small inter fibrillar spaces or pores. Bar = 10 µm (**F**).

**Figure 5 biomedicines-11-01788-f005:**
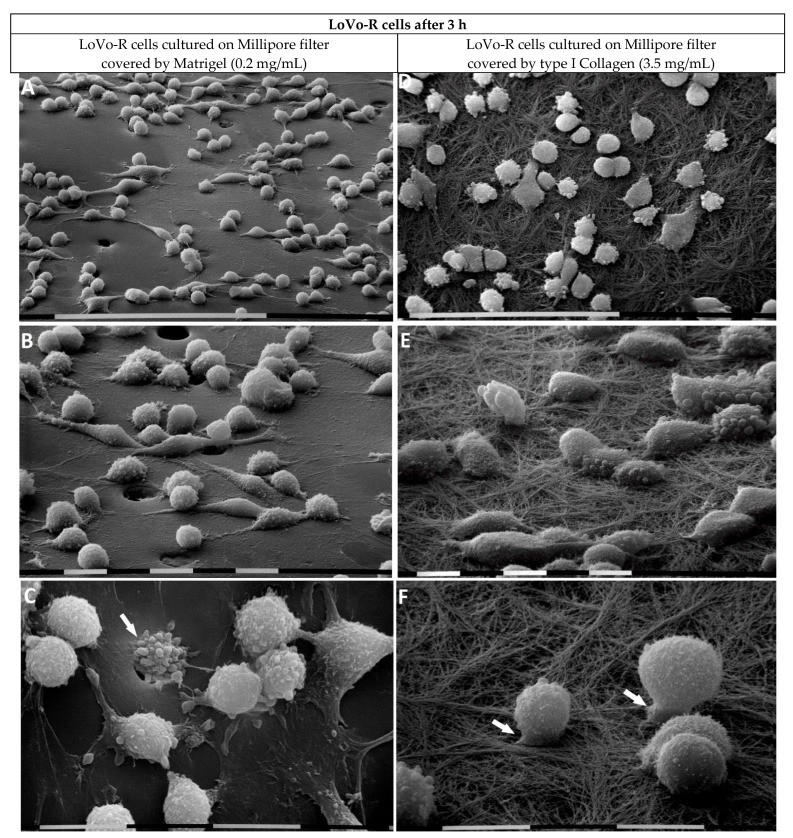
LoVo-R cells cultivated for 3 h on Millipore filter with 8 µm pores covered by Matrigel (0.2 µg/µL) mimicking the BM or type I Collagen (3.5 µg/µL) reproducing the collagen meshwork of a desmoplastic lamina propria. LoVo-R cells growing on Matrigel are more isolated than the LoVo-S cells (see Figure 3A). Bar = 100 µm (**A**). LoVo-R cells primarily show a globular shape with protruding exosomes and microvesicles on the surface, but few isolated elongated-fusiform mesenchymal-shaped cells are detectable. Bar = 10 µm (**B**). Globular-shaped cells developing extravesicles. A cell passing through a Millipore pore (arrow). Bar = 10 µm (**C**). LoVo-R cells cultivated on type I Collagen meshwork are isolated and do not show cell–cell contact. Bar = 100 µm (**D**). All the cells, adhering to the collagen fibrils, exhibit a rounded or globular shape and develop exosomes and microvesicles. Bar = 10 µm (**E**). Isolated globular-shaped cells develop cytoplasmic protrusions or short filopodia, which penetrate into the collagen layer (arrows). Bar = 10 µm (**F**).

**Figure 6 biomedicines-11-01788-f006:**
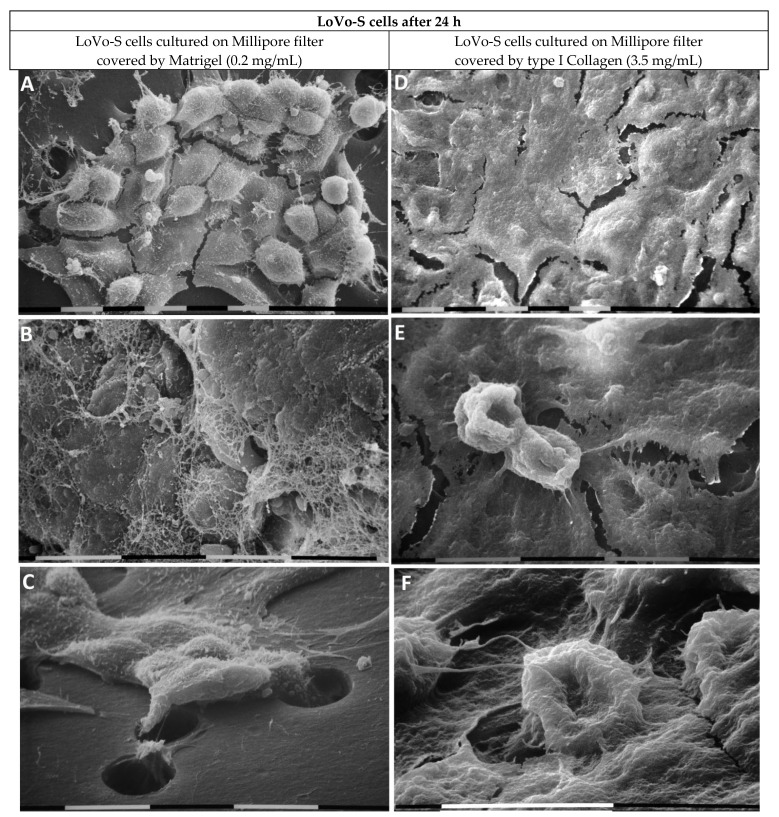
LoVo-S cells cultivated for 24 h on Millipore filter with 8 µm pores covered by Matrigel (0.2 µg/µL) mimicking the BM or type I collagen (3.5 µg/µL) mimicking the collagen network of the desmoplastic lamina propria. The LoVo-S cells cultivated on Matrigel appear as grouped flattened epithelial cells exhibiting tight contact with each other (**A**). bar = 10 µm. The cells are surrounded by a fibrillar meshwork likely corresponding to remnants of the Matrigel substrate. Bar = 10 µm (**B**). Some grouped LoVo-S cells are driven by a leader cell, which develops short filopodia to cross the pores of the Millipore filter. Bar = 10 µm (**C**). The LoVo-S cells cultivated on type I Collagen show an epithelial phenotype. They appear as polygonal and flattened cells with tight cell–cell contacts. Bar = 10 µm (**D**). Two funnel-shaped cells are detectable on the continuous layer of the flattened cells. Bar = 10 µm (**E**). A single invaginating funnel-shaped cell invades the Matrigel and penetrates a pore of the Millipore filter. Bar = 10 µm (**F**).

**Figure 7 biomedicines-11-01788-f007:**
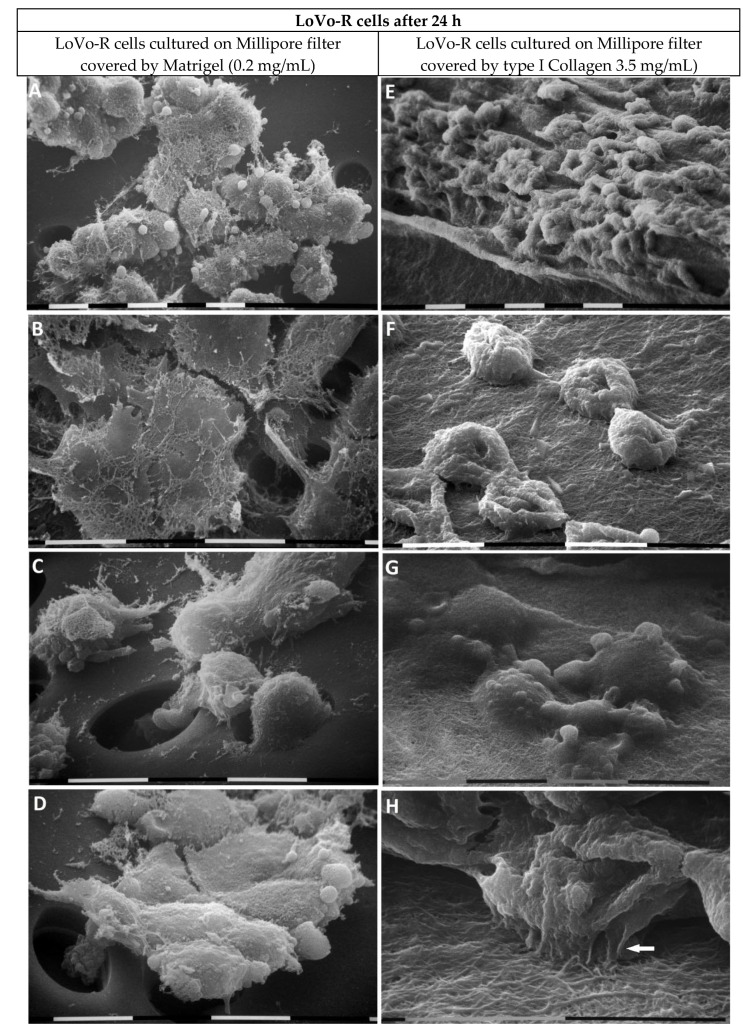
LoVo-R cells cultivated for 24 h on Millipore filter with 8 µm pores covered by Matrigel (0.2 µg/µL) mimicking the BM or type I Collagen (3.5 µg/µL) reproducing the collagen array of the desmoplastic lamina propria. The LoVo-R cells cultivated for 24 h on the Matrigel layer appear as grouped epithelial cells with cell–cell contact but developed more microvilli and extravesicles on their surface compared to the LoVo-S cells (see Figure 5A). Bar = 10 µm (**A**). Remnants of Matrigel envelope the LoVo-R flattened cells. Bar = 10 µm (**B**). Single cells penetrate the pores of the Millipore filter. Bar = 10 µm (**C**). Clusters of grouped LoVo-R cells are driven by a leader cell, which developed filopodia and crosses a pore of the Millipore filter. Bar = 10 µm (**D**). The LoVo-R cells cultivated on type I Collagen meshwork show a funnel-shaped phenotype and cell-cell contact through thin and short filopodia. Bar = 10 µm (**E**,**F**). Isolated cells developing cytoplasmic extravesicles are completely covered by fibrils, suggesting LoVo-R cells are invaginating the collagen layer. Bar = 10 µm (**G**). At higher enlargement, the LoVo-R cells develop ventral thin cytoplasmic protrusions, which morphologically correspond to invadopodia (arrow) and adhere to the collagen fibril layer. Bar = 10 µm (**H**).

**Figure 8 biomedicines-11-01788-f008:**
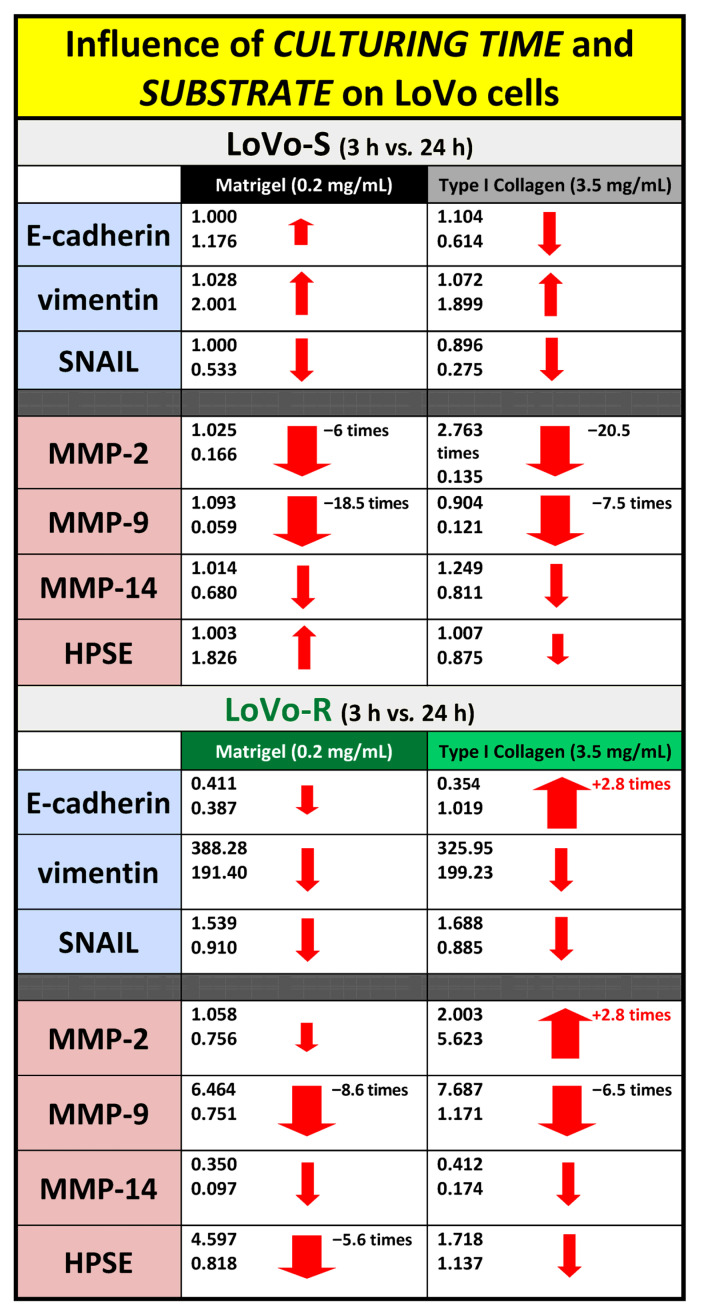
Different roles of time of culturing and substrate in MMPs gene expression in LoVo-S and LoVo-R cultivated on Matrigel or type I Collagen meshwork after 3 and 24 h. Over time, LoVo-S cultured on type I collagen showed a decrease in E-cadherin, SNAIL, MMP-2, -9, and -14 expression and an increase in vimentin. The decrease in MMP-2 expression seems to be related to duration but is particularly influenced by type I Collagen substrate. The LoVo-R cells cultured on the same substrate exhibited a decrease in SNAIL, MMP-9, -14, and HPSE expression but an increase in E-cadherin and MMP-2 expression, which were also related to the type of substrate. The size of the arrows is approximately related to the range of increase/decrease.

**Figure 9 biomedicines-11-01788-f009:**
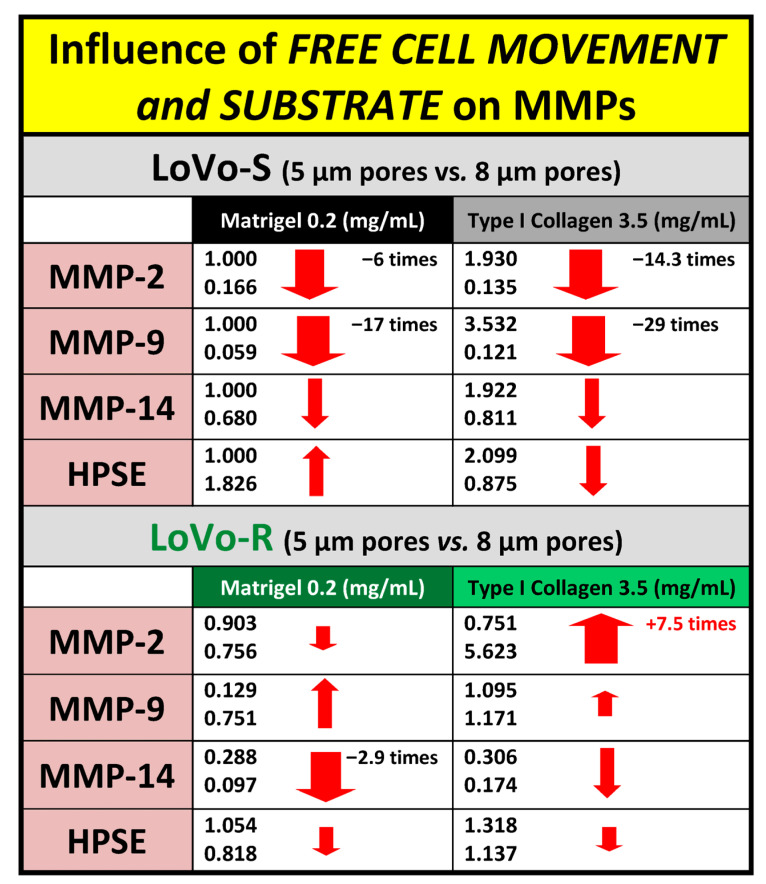
Different roles of free cell movement in MMPs gene expression in LoVo-S and LoVo-R cultivated on Matrigel or type I Collagen meshwork, covering Millipore filters not allowing cell crossing or Millipore filters allowing cell migration through the pores after 24 h. As time passes, in cultures allowing free cell movement, type I Collagen strongly reduces MMP-2 and -9 expression in LoVo-S cells. In contrast, the same substrate highly increases MMP-2 expression in LoVo-R cells. The size of the arrows is approximately related to the range of increase/decrease.

**Table 1 biomedicines-11-01788-t001:** List of real-time qPCR primers used in this study.

Gene	Primer Sequence
*E-Cadherin*	F: TTCTGCTGCTCTTGCTGTTT, R: TGGCTCAAGTCAAAGTCCTG;
*Vimentin (VIM)*	F: AAAACACCCTGCAATCTTTCAGA, R: CACTTTGCGTTCAAGGTCAAGAC;
*SNAIL*	F: AGTTTACCTTCCAGCAGCCCTAC, R: AGCCTTTCCCACTGTCCTCATC;
*MMP-2*	F: TGCATCCAGACTTCCTCAGGC, R: TCCTGGCAATCCCTTTGTATGTT;
*MMP-9*	F: GGTGATTGACGACGCCTTTG, R-CTGTACACGCGAGTGAAGGT;
*MMP-14*	F: TGCCATGCAGAAGTTTTACGG, R: TCCTTCGAACATTGGCCTTG;
*Heparanase (HPSE)*	F: ATTTGAATGGACGGACTGC R: GTTTCTCCTAACCAGACCTTC;
*GAPDH*	F: ACACCCACTCCTCCACCTTT R: TCCACCACCCTGTTGCTGTA;

## Data Availability

The original contributions presented in the study are included in the article material. Further inquiries can be directed to the corresponding author.

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
