# Peer review of "Colorectal Cancer Cell Invasion and Functional Properties Depend on Peri-Tumoral Extracellular Matrix"

_biomedicines, 2023, doi:10.3390/biomedicines11071788_

Round 1
Reviewer 1 Report
The authors are commended for their excellent SEM work. The SEM images demonstrated in this study were really outstanding.
Minor comments:
1. This study would have been much stronger if the authors could include other CRC cell lines. CRC cells are known to contain different mutations and each mutation contribute to the different phenotype of the cells. Based on just LoVo cells, it is really difficult to rule out the possibility of cell line specific phenomenon.
2. On Page 8, lines 225 to 229, the authors stated “LoVo-S cells cultured on Millipore filter covered by Matrigel (0.2 mg/ml) appeared grouped and all of them showed cell-cell contacts. To probably explore the surrounding microenvironment they developed long filopodia and intercellular tunneling nanotubes. Cells mainly showed a globular shape developing extravesicles on their cytoplasmic surface, but very few elongated- mesenchymal shaped ones were also detectable (Figs. 4 A-C).”
It is of my interest to know the nature of these extravesicles on the cytoplasmic surface. Could the authors elaborate more on this finding?
3. Continued from 3, have the authors performed any biochemical tests to identify the molecular nature of these extravesicles? Could these be exosomes? Or membrane invaginations prior to cell death?
4. It would be interesting to see if adding inhibitor of type I collagen could reverse the phenomena observed in this study.
Author Response
Reviewer 1
Comments and Suggestions for Authors
The authors are commended for their excellent SEM work. The SEM images demonstrated in this study were really outstanding.
We thank very much the Reviewer for his/her the compliments, hoping that also readers could appreciate the pictures as well as the biomolecular results.
Minor comments:
- This study would have been much stronger if the authors could include other CRC cell lines. CRC cells are known to contain different mutations and each mutation contribute to the different phenotype of the cells. Based on just LoVo cells, it is really difficult to rule out the possibility of cell line specific phenomenon.
Yes it is true. We also thought to include other CRC cell lines in our project but the collected data would have been heavy in a very long paper. Anyway, we are just preparing further investigations involving other CRC cells which will be compared with data concerning the LoVo cells. Indeed, in previous investigations the LoVo-S and LoVo-R cells were quite different, so that they can be functionally considered as two different cell lines. However, they have a common origin, so that they seemed us ideal to investigate the different behavior of drug sensitive cells then becoming drug resistant.
- On Page 8, lines 225 to 229, the authors stated “LoVo-S cells cultured on Millipore filter covered by Matrigel (0.2 mg/ml) appeared grouped and all of them showed cell-cell contacts. To probably explore the surrounding microenvironment they developed long filopodia and intercellular tunneling nanotubes. Cells mainly showed a globular shape developing extravesicles on their cytoplasmic surface, but very few elongated- mesenchymal shaped ones were also detectable (Figs. 4 A-C).”
It is of my interest to know the nature of these extravesicles on the cytoplasmic surface. Could the authors elaborate more on this finding?
We completed a sentence to add more explanations in the Discussion: “They (referring to MMPs) are secreted by exocytosis from cytoplasmic protrusions called invadopodia, but may be also released in the TME from cytoplasmic extravesicles shed by a superficial blebbing of the plasma membrane. Extravesicles include multivesicular bodies or exosomes (50–200 nm), vesicles deriving from direct budding of the plasma membranes or microvesicles (50–1000 nm) and larger vesicles or apoptotic bodies (>1000 nm). They contain lipids, second messengers, genetic material, as well as HPSE [32, 33].”
- Continued from 3, have the authors performed any biochemical tests to identify the molecular nature of these extravesicles? Could these be exosomes? Or membrane invaginations prior to cell death?
Thank you for your question which allows us to explain what we reported in the results. We did not perform any biochemical test to identify the molecular nature of the observed extravesicles because we could not isolate them, but their size allowed us to identify most of them, as we specified in the text. Following a morphological classification, the extravesicles have been divided in three group, just, considering their size: the vesicles originating via exocytosis of multivesicular bodies are called exosomes (50–200 nm), while vesicles derived from direct budding of the plasma membranes are called microvesicles (50–1000 nm) and vesicles released during apoptosis are called apoptotic bodies (> 5 μm). Therefore, considering the size of the observed extravesicles which were mostly present in LoVo-R cells (Figs. 5), they must be classified as exosomes and microvesicles.
- It would be interesting to see if adding inhibitor of type I collagen could reverse the phenomena observed in this study.
We thank the Reviewer for the very intriguing suggestion. We aim to start a project involving co-cultures of different CRC cells and fibroblasts to study the effect of adding an inhibitor of type I collagen.

Reviewer 2 Report
In this study, the Authors investigated how the extracellular matrix
affects colorectal cancer cell behaviour during a spatiotemporal
invasion. Using methods of RT PCR, western blot and scanning
electron microscopy, the Authors indicated that type 1 collagen meshwork
attenuates agressiveness of drug sensitive cells and improves a proteolytic
in drug resistant cells. This article provides important evidence and can be
published. There are only small concerns:
Abstract: Please, provide a background describing the importance of the
study. 2. It also would be good to srtucture and concentrate Results
and to provide conclusion explaining medical application of the obtained
results.
English is quite good.
There are only several small typos.
Please, re read the text several times.
Author Response
Reviewer 2
Comments and Suggestions for Authors
In this study, the Authors investigated how the extracellular matrix affects colorectal cancer cell behaviour during a spatiotemporal invasion. Using methods of RT PCR, western blot and scanning electron microscopy, the Authors indicated that type 1 collagen meshwork attenuates agressiveness of drug sensitive cells and improves a proteolytic in drug resistant cells. This article provides important evidence and can be published.
There are only small concerns:
- Abstract: Please, provide a background describing the importance of the study.
We thank the Reviewer for remembering us this lack. We added a sentence to Abstract:
“Experimental studies on CRC cells should examine the peri-tumoral ECM components, as well as the dynamic physical conditions of TME which differently affect the behavior and aggressiveness of both drug-sensitive and drug-resistant LoVo cells.”
- It also would be good to structure and concentrate Results and to provide conclusion explaining medical application of the obtained results.
The many data regarding two cells types cultivated in two substrates at two different times make us difficult to reduce the Results section which was simplified by creating different short paragraphs. Following the Reviewer suggestion, we revised the Conclusions section and added a concluding sentence explaining possible medical applications of the obtained results:
“Understanding how some ECM components affect the drug resistant CRC cells will help upcoming therapeutic procedures aiming to modify the altered ECM of the TME.”
Round 2
Reviewer 1 Report
The authors have provided sufficient responses to all the issues I had with the first version of the manuscript. I do not have further questions.